# Buffalo Yogurt Fortified with Eucalyptus (*Eucalyptus camaldulensis*) and Myrrh (*Commiphora Myrrha*) Essential Oils: New Insights into the Functional Properties and Extended Shelf Life

**DOI:** 10.3390/molecules26226853

**Published:** 2021-11-13

**Authors:** Ahmed Mohamed Hamed, Awad A. Awad, Ahmed E. Abdel-Mobdy, Abdulhakeem Alzahrani, Ahmad Mohammad Salamatullah

**Affiliations:** 1Dairy Science Department, Faculty of Agriculture, Cairo University, Giza 12613, Egypt; awad.awad@agr.cu.edu.eg (A.A.A.); Dr.Ahmed.Emam@cu.edu.eg (A.E.A.-M.); 2Department of Food Science & Nutrition, College of Food and Agricultural Sciences, King Saud University, 11 P.O. Box 2460, Riyadh 11451, Saudi Arabia; aabdulhakeem@ksu.edu.sa (A.A.); asalamh@ksu.edu.sa (A.M.S.)

**Keywords:** functional yogurt, *Eucalyptus* oil, Myrrh oil, antioxidant activity, antibacterial activity

## Abstract

*Eucalyptus* (*Eucalyptus camaldulensis*) and Myrrh (*Commiphora Myrrha*) essential oils (EOs) stand out for their benefits in terms of health and functionality. Buffalo set yogurt enriched with different concentrations of EOs (0.3, 0.6, and 0.9%) were investigated. The effects of addition on sensory, syneresis, antibacterial activity, and bioactive properties (total phenol content and antioxidant activity) of yogurt were studied. The most acceptable organoleptic properties of treated yogurt were those samples treated with *Eucalyptus* oil. The levels of syneresis were decreased by increasing the concentration of EOs. Moreover, the antioxidant activity, antibacterial activity, and total phenolic content were enhanced by increasing the concentration of EOs. Yogurt with 0.9% *Eucalyptus* oil showed the highest antioxidant activity and total phenolic content. The same concentration of Eucalyptus oil showed the highest antibacterial activity against *S. typhimurium* (the inhibition zone was 20.63 mm) then *E. coli* (the inhibition zone was 19.43 mm). On the other hand, the highest antibacterial effect against *L. monocytogene* was for Myrrh oil-enriched yogurt by 0.9% and the inhibition zone was 19.21 mm. The obtained results showed that *Eucalyptus* and Myrrh oils can be applied to yogurt to improve its beneficial properties in terms of physical characteristics and for human health due to their antioxidant activity and phenolic materials.

## 1. Introduction

Functional yogurt plays an important role in human nutrition due to its content of proteins, lactose, calcium, and water-soluble vitamins [1]. In general, milk and its derivatives are not seen as a rich source of phenolic components [2]. Over the last few years, some functional milk products have been produced using fortification and enrichment, as consumers need good tasting nutritious foods [3]. Yogurt has thus begun to draw new customer groups because of its good taste and enhanced health benefits [4]. To enhance the quality of newly designed functional food products, which are foods that have a potentially positive effect on health beyond basic nutrition, food fortification is defined by the inclusion of one or more components regardless of their natural presence in food (since they occur naturally in food products) [5]. Essential oils (EOs) are often used in the food industry and are used for flavor, aromatherapy, additives, biopesticides, and pharmaceuticals because of their strong bioactivity, aromatic features, and many biological activities [6]. Chemical preservatives cannot eliminate several pathogenic bacteria, such as *Listeria monocytogenes* (Murray et al., 1926; Pirie, 1940), in food products or delay the growth of spoilage microorganisms [7]. Myrrh is a wholesome, natural flavors product approved by the US Administration for Food and Drugs (FDA, 2018; 21CFR172.510). *Commiphora Myrrha* (T. Nees) Engl. (common name: Myrrh) comes from Burseraceae family [8]. Extracts from various plant types were used as flavoring and seasoning agents in foodstuffs and beverages and as food preservatives in ancient times. Myrrh adds a typical taste and its antioxidant, bacteriostatic, and bactericidal effects enhance food storage [9]. It is also antibacterial and is used for many conditions. It also demonstrates significant inhibitory activity against *Salmonella* entericaserovar typhimirium [10]. Because sesquiterpenes are present in Myrrh, it possesses antibacterial and antimicrobial effects. Michie and Cooper [11] reported that in 1100bc Myrrh was used for the first time in the treatment of infected teeth and gut worms by Sumerians [12]. The Egyptians used Myrrh in ancient times to embalm. Myrrh oil has been used for several years to treat skin injuries and fungal infections caused by Candida albicans and Tineapedis [13]. Myrrh is an herbal product that has been used since ancient times for traditional medication and other purposes and it is safe to use and not poisonous to people [14]. In addition, it is generally recognized as a safe (GRAS) assessment program of the Flavor and Extract Manufacturers Association (FEMA) of the United States [15]. Different species of genus *Eucalyptus* L’Hér oil are used in the food, toiletries, pharmaceuticals, and cosmetics industries, and these applications are due to the anti-inflammatory, antioxidant antiseptic, flavoring, and antihyperglycemic properties of the molecules that are present in the oil [16]. A few studies investigated their activity against pathogenic and food spoilage bacteria [17]. *Eucalyptus* oil in Western civilization has been used as an expectorant for colds and coughs as well as a disinfectant for skin, myalgias liniment, and a counterirritant [18]. It has been found that the activity of antimicrobial *Eucalyptus* oils can differ significantly within microbial strains and species [19]. In a natural habitat, the buffalo grazing is a determinant of soil and vegetation that can affect the structure and function of vegetation coverage in many ways [20], for which it is necessary to establish a grazing plan [21,22] and crucial management actions which have effects on the antioxidant activity and chemical composition of the final product, such as yogurt and milk, while a confined environment plays a crucial role in the food provided by the farmer. Industrial milk processing is supplied by both big and small intensive farms working with cattle from various origins: specialized dairy breeds under advantageous environmental conditions, such as irrigated schemes with consistent green fodder supply [23]. To the best of our knowledge, this is the first work applying buffalo yogurt using both *Eucalyptus* and Myrrh oils to improve the physical characteristics due to their antioxidant activity and to extend the shelf life of the resultant yogurt. The study’s main goal was to examine the influence of *Eucalyptus* and Myrrh oil in various ratios of the yogurt’s functional properties. The impact of these concentrations on the antimicrobial, chemical, physical and sensory properties of the resultant yogurt was investigated as well as the antioxidant activity.

## 2. Materials and Methods

### 2.1. Materials

Fresh buffalo milk used in yogurt manufacturing was obtained from the herd of the Faculty of Agriculture, Cairo University, Giza, Egypt. The starter cultures starter culture consists of a 50:50 mixture of *Lactobacillus delbrueckii subsp. bulgaricus* and *Streptococcus salivarius subsp. Thermophilus* were obtained from Chr. Hansen, Hoersholm, Denmark. *Eucalyptus* and Myrrh essential oils were purchased from the Primagricu company, Benisuef, Egypt.

### 2.2. Methods

#### 2.2.1. GC-MS Identification of Eucalyptus and Myrrh Oils Individual Compounds

The identification of *Eucalyptus* and Myrrh oil compounds was carried out using gas chromatography (Agilent Hewlett-Packard 6890, Agilent Technologies, Santa Clara, CA, USA) coupled with a mass spectrometer detector (MSD) (Agilent Technologies 5973N, Santa Clara, CA, USA). The method of identification was according to that described in [24,25]. To determine the compounds, Spectra was used for the identity of unknown compounds with the National Standards Institute of Technology (NIST, 98). Relative percentage amounts from GC-MS were measured using a commercial MS library (NIST98). The percentage values were the average of three sample injections.

#### 2.2.2. Yogurt Manufacturing

The homogenized buffalo milk used had 6.1% fat, 3.5 ± 0.3% protein, and 84.3 ± 0.2% moisture. Homogenization of all the batches was carried out at 60 °C in a two-stage homogenizer (M/S Goma Engineers, Mumbai, India) with 1000 and 500 psi pressures at the first and second stages, respectively. The milk was heat-treated at 85 °C for 10 min, then cooled to 45 °C. The milk was divided into seven equal parts (C, E1, E2, E3, M1, M2, and M3). The first part was a control (C) with no EOs oil, and the three treatments were generated with various concentrations of *Eucalyptus* oil (0.3%, 0.6%, and 0.9%) referred to as E1, E2, and E3, respectively, and the other three treatments were generated with various concentrations of Myrrh oil (0.3%, 0.6%, and 0.9%) referred to as M1, M2, and M3, respectively, which were added before homogenization. These three concentrations were used after manufacturing the different formulations of yogurt and a preliminary sensory evaluation study was conducted to select the best formulations that will be used throughout the experiment. It was then inoculated into the processing vat at a concentration of 2% with yogurt starter cultures (50:50 mixture of *Lactobacillus delbrueckii subsp. bulgaricus* and *Streptococcus salivarius subsp. Thermophilus*). Starter cultures were prepared by cultivation in 10% nonfat dry milk, autoclaved at 115 °C for 15 min, inoculated under sterile conditions, incubated at 42 °C, and kept in a refrigerator at 4 °C until use. Then, the inoculated milk was incubated at 42 °C. Once the pH reached 4.6, which was controlled by sampling either in the fermentation tanks or directly in cups, the yogurt was packaged in covered plastic cups and was cooled down to 4–5 °C.

#### 2.2.3. Physical and Chemical Analysis of Yogurt

Dry matter and fat content were calculated using official AOAC [26] techniques in samples of milk and various milk products. A laboratory pH meter was used to read the pH values (HI 93 1400, Hanna instruments, Smithfield, RI, USA). All measurements were carried out in triplicate.

#### 2.2.4. Determination of Whey Syneresis

A 100 mL yogurt sample was weighed on a filter paper placed over a funnel, which measured Syneresis’ susceptibility to yogurt. The quantity of the whey collected in a beaker was calculated following 1 h of drainage and was used as a syneresis index. [27]. For the measurement of syneresis, the following formula was used:Syneresis % = (W_1_ × 100)/W_2_
where: W_1_ = weight of whey collected after drainage; W_2_ = weight of yogurt sample. All measurements were carried out in triplicate. 

#### 2.2.5. Sensory Evaluation

A random selection of yogurt samples was evaluated for availability, surface smoothness, mouthfeel, and texture (soft, grain, springiness), flavor, and taste (sweetness, aroma), and overall admissibility by members of the department of dairy science of the faculty of agriculture at Cairo University. All these trained panelists (15 people) were experts and were chosen based on their desire to participate and their knowledge about dairy products. They were frequent yogurt consumers and did not have any allergies to it. Plain- and fortified-yogurt samples were presented in plastic coded (three-digit random codes) cups. Each cup contains 50 mL of yogurt samples that were freshly removed from the refrigerator. According to [28], the yogurt samples were evaluated on a nine-point scale for appearance, mouth feels, texture, flavor, and overall acceptability.

#### 2.2.6. Total Phenolic Content

The microscale Folin–Ciocalteau method was used to calculate total phenolic content [29]. Gallic acid standards, samples (1.58 mL), or blanks of water were mixed in the cuvette and allowed to stand at room temperature for 8 min with 100 μL Folin–Ciocalteau reagent. A 20% Na2CO3 (200 μL) solution was added. Sample and standard absorption were tested at 765 nm. The gallic acid equivalents based upon a gallic acid standard curve were measured as total phenolic using a UV/Vis. Spectrophotometer (Jenway, Staffordshire, ST15 OSA, UK).

#### 2.2.7. Radical Scavenging Activity

Radical scavenging activity was carried out according to [30,31]. In brief, water-soluble extract (WSE) was prepared in 0.1 M sodium phosphate buffer, pH 7.0, containing 1% (wt/vol) Triton X-100, and 2,2-diphenyl-1-picrylhydrazyl (DPPH; 100 μM) was prepared in organic solvent (methanol). An aliquot (1.5 mL) of WSE (sample) or 1.5 mL of buffer (control) was mixed with 1.5 mL of DPPH solution and was left in the recommended condition in the dark at room temperature for 30 min. Then the absorption of the solution by spectrophotometer was measured at 517 nm. The decreasing percentage of absorbance of the sample relative to the control was calculated as the relative scavenging activity [32].

#### 2.2.8. Antibacterial Activity

To obtain the yogurt supernatant, samples were centrifuged (Centrifuge version C-28 AC BOECO, Hamburg, Germany) at 4000× *g* for 30 min at 4 °C [29]. The supernatants obtained were filtered with the 0.45-μm diameter millipore membrane filter and were stored at −20 °C for further analysis (antibacterial activity). 

The supernatant filters were used to assess the antibacterial efficacy of certain pathogens (*E. coli ATCC35218*, *L. monocytogenes ATCC19115*, and *S. typhimurium ATCC14028*) according to [29] using an agar well diffusion assay. A diluted sample of 100 μL of yogurt supernatant was distributed on agar plates. The inhibition zone was measured after incubation at 37 °C/24–48 h. The bacterial activity was expressed as the inhibition zone (mm). Each test was conducted three times.

#### 2.2.9. Statistical Analysis

The obtained data were statistically analyzed using two-way ANOVA using MSTAT-C software to assess the significant differences (*p* < 0.05) between the means of samples and the storage period. All data were presented as a mean ± standard deviation of three replicates. The means of results were compared by the Tukey test with a confidence interval set at 95%.

## 3. Results and Discussions

### 3.1. GC-MS of EOs

GC/MS analysis of *Eucalyptus* and Myrrh oils allowed the identification of different compounds (Table 1). For Myrrh oil, the analysis allowed the identification of 12 compounds reflecting 99.99% of the essential oil. The major compounds identified in Myrrh oil are presented in Table 1. Results indicate a significant amount of the major constituents identified in the Myrrh oil (10.6%, 7.4%, 18.4%, 23.4%, and 6.2%) for elements, Germacrene B and D, isofuranogermacrene, Furanodieneb, Furanoeudesma-1,3-diene, Lindestrene, and 2-acetoxyfuranodiene., respectively. Some previous studies reported by [33] found that the major compounds identified in Myrrh resin were 2-acetoxyfuranodiene (9.80%), furanoeudesma-,3-diene (8.97%), isofuranoger macrene (6.71%), and epicurzerenone (3.643%), followed by 2- methoxyfuranodiene (2.97%) and lindestrene (2.74%). In the same context, its main components, identified and quantified by GC/MS, were furanoeudesma-1,3-diene, 34.9%; lindestrene, 12.9%; curzerene, 8.5%; and germacrone, 5.8% [34]. It was noted that there are some differences, whether in the proportion of Myrrh compounds or their types, this could be due to the type of plant or the environment. For *Eucalyptus* oil, the major chemical composition of the *Eucalyptus* camaldulensis essential oil grown in northern Egypt is α- pinene (24.83%), 1,8-cineol (47.8%), Terpineol alpha (5.55%), pinocarveol (3.42%), 4-terpineol (1.66%), Trans-Pinocarveol (7.34%) and Pinacarvone (4.11%).

### 3.2. Sensory Evaluation

Table 2 shows the average sensory assessment scores for plain and fortified yogurt with Myrrh oil on days 1 and 14 of refrigerated storage. In terms of sensory characteristics, the yogurt samples vary in appearance, mouthfeel, texture, flavor, and overall acceptability. The plain sample had the best sensory qualities on the first day. However, at the end of storage (day 14) the control group’s scores for mouthfeel, appearance, texture, and overall acceptability decreased. All the sensory properties of yogurt fortified with Myrrh oil decreased after 14 days of refrigerated storage at 4 °C, which is likely due to a drop in aromatic components in yogurt. However, yogurt fortified with *Eucalyptus* oil was more acceptable than yogurt fortified with Myrrh oil. Other yogurt types remained unchanged. Other sensory indices (plain yogurt, yogurt fortified EOs) decreased with storage to day 14, but not significantly (*p* < 0.05) compared to day 1. According to [35], the acetaldehyde content of yogurt decreases throughout the storage period. This lower score may be due to an increase in the acidity of the yogurt, which could prevent the release of aromatic components [36]. The results showed that storage time influenced yogurt sensory characteristics and that all sensory attributes decreased in all types. In addition, yogurt fortified with *Eucalyptus* oil has more acceptability than yogurt fortified with Myrrh oil.

### 3.3. Physicochemical Analysis

The findings from the chemical analysis of yogurt samples are shown in Table 3. It was observed that there were no significant differences due to the addition of oils in different concentrations to the total solids (TS) of yogurt samples. In addition, there are no big differences between it and the control. Dhawi et al. [29] stated that the mass fraction of total dry matter in yogurt samples with plant extract was lower than that without extract. In this study, the TS content of EOs-enriched yogurt samples was found to be higher than regulated, whereas when fortified with 0.9% Eos, the highest dry matter rate was reported for both types of oils. In addition, these are aligned with the findings of [37], who reported that yogurt supplemented with peanut skin extract has a higher percentage of TS. Regarding the pH of yogurt samples, it decreased linearly during the storage period in yogurt samples. EOs-enriched yogurt samples containing 0.9% exhibited significantly (*p* < 0.05) higher pH than the other samples and yogurt enriched with *Eucalyptus* oil has the highest value of 4.70. It was found that the pH ranged from 4.67 to 4.70 in all treatments of yogurt fortified with peanut skin and, during cold storage, the pH values are decreased in all treatments to reach 4.42 with significant differences between treatments [38].

### 3.4. Whey Syneresis

Table 3 summarizes the syneresis percentage of yogurt. For the control samples, the syneresis (%) of the yogurts after 4 °C ranged from 9.84% to 12.43%. However, by increasing the concentration of Eos, the syneresis percentage was decreased. Syneresis percentage ranged between 8.44% and 10.83% for *Eucalyptus* oil and ranged between 8.12% and 11.33% for Myrrh oil. Syneresis% was increased by storage period in all types of yogurts. Throughout the storage period, the percentage of wheying-off was higher in control samples than in other treatments. The propensity to syneresis was lower in yogurt fortified with various concentrations of Myrrh oil (0.9%) than in control yogurt and yogurt fortified with *Eucalyptus* oil. The lower syneresis of yogurt fortified with Myrrh oil may be explained by the higher solid content (Table 3). Low-solid yogurts have a higher degree of syneresis than high-solid yogurts [36,37,39].

### 3.5. Total Phenolic Content (TPC)

Table 4 summarizes the TPC in water extracts of control and yogurt fortified with various concentrations of EOs. According to the current findings, the highest phenolic content was found in yogurt fortified with different concentrations of EOs, accompanied by lower concentrations of Myrrh oil, and finally the control sample. TPC levels in yogurt fortified with EOs before bacterial fermentation was higher than in the control sample. However, after two weeks of storage, TPC decreased in all kinds of yogurt to be 5.12, 15.45, 23.58, 26.35, 14.31, 22.63, and 25.33 for control—E1, E2, E3, M1, M2, and M3, respectively. As previously stated in previous studies, this was linked to the milk polyphenol interaction [40]. The formation and degradation of polyphenolic compounds were observed during the fermentation of milk by yogurt bacteria [41].

### 3.6. Antioxidant Activity

DPPH methods have been used to assess antioxidant activity in this analysis. Fresh control yogurt showed the lowest antioxidant activity value (Table 4) as radical scavenging activity (RSA%). Adding EOs resulted in a higher RSA percentage ratio, which depends on the concentration. Yogurt with 0.9% oil showed the highest RSA percent values (75.58% and 71.21% for E3 and M3, respectively) in fresh samples, while the control yogurt had the least value (15.33%) on day 1 of storage. With cold storage of control samples and fortified treatments (E1, E2, E3, M1, M2, and M3), the RSA percent decreased even with the highest RSA percentage values to be 13.82%, 45.11%, 56.84%, 67.45%, 43.72%, 53.21%, and 63.21% for control, E1, E2, E3, M1, M2, and M3, respectively). It is due to their hydrogen-donating or electronic potential that the antioxidant affects radical DPPH scavenging [42]. All EOs-rich yogurts were found to display an improvement in the percentage of RSA compared to plain yogurt, and the concentration of *Eucalyptus* oil has more RSA than Myrrh oil. *Eucalyptus* oil has been used in pharmaceutical products and traditional medicine and is also used as a flavoring agent in food products due to its desirable flavor, antioxidant activities, and antimicrobial activities [43]. Data from [44,45] showed that fortified yogurt has been used to investigate the function of antioxidants with the radical scavenging system of DPPH and its different types of natural ingredients. When compared to plain yogurt, all kinds of EOs-enriched yogurts display a higher antioxidant activity.

### 3.7. Antibacterial Activity

The antibacterial activity was measured using the agar well diffusion method, and the results were expressed as inhibition zone diameter (mm) (Figure 1). Antibacterial activity of yogurt enriched with EOs was significantly higher (*p* < 0.05) against *E. coli*, *Salmonella typhimurium, and Listeria monocytogene*. Yogurt enriched with 0.9% *Eucalyptus* oil showed the highest antibacterial activity against *S. typhimurium* (the inhibition zone was 20.63 mm) then *E. coli* (the inhibition zone was 19.43 mm). On the other hand, the highest antibacterial effect against *L. monocytogene* was for M3 and the inhibition zone was 19.21 mm. Essential oils extracted from different plants had antibacterial and antifungal properties. In the same way, [46] reported that significant growth-inhibiting effects on Gram-positive (*E. coli*) and Gram-negative bacteria (*S. aureus*) have been observed because of *Eucalyptus* oil. Reference [47] mentioned that the addition of 1.0 g/L from anise essential oil and oleoresin is influential in controlling the growth of spoilage microorganisms in all samples of yogurt. Finally, [48] found that the essential oils were effective in inhibiting the growth of spoilage microorganisms in yogurt. This could be due to the addition of *Eucalyptus* and Myrrh oil, which had antibacterial activities versus a wide range of mold, yeast, and pathogenic bacteria.

## 4. Conclusions

Fortifying yogurt with essential oils is of great interest to improve its functionality and increase its shelf life. According to the obtained data, *Eucalyptus* and Myrrh oil can be used successfully in the manufacturing of yogurt. The additions of 0.3%, 0.6%, and 0.9% of EOs levels to yogurt improved its physicochemical properties and extended its shelf life with appropriate acceptability and sensory properties. Acidity and the lack of growth of pathogenic microorganisms were considered the significant factors influencing the acceptability and shelf life of yogurt. Furthermore, fortified yogurt with *Eucalyptus* and Myrrh oil can be considered a novel product with functional healthy properties. Certainly, the *Eucalyptus* and Myrrh oil can be considered a preservative agent used in yogurt to extend its shelf life while avoiding using chemical preservatives.

## Figures and Tables

**Figure 1 molecules-26-06853-f001:**
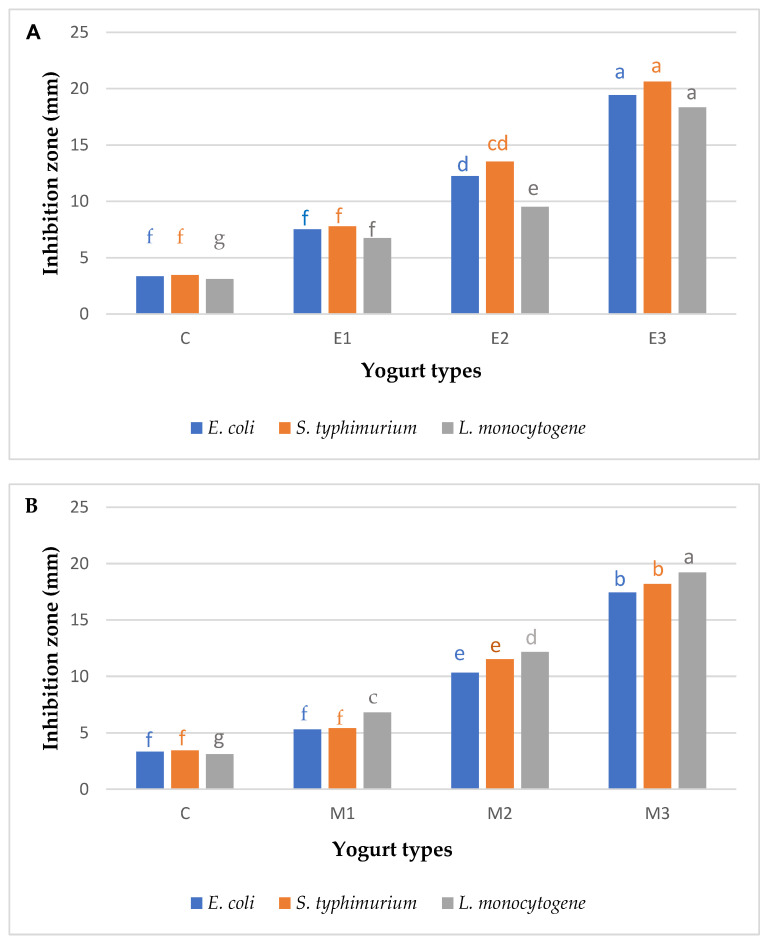
Antibacterial activity of plain yogurts and yogurts supplemented with *Eucalyptus* (**A**) and Myrrh (**B**) oil during cold storage. C: plain yogurt, E1: yogurt fortified with 0.3% *Eucalyptus* oil, E2: yogurt fortified with 0.6%, *Eucalyptus* oil, E3: yogurt fortified with 0.9% *Eucalyptus* oil, M1: yogurt fortified with 0.3% Myrrh oil, M2: yogurt fortified with 0.6%, Myrrh oil, M3: yogurt fortified with 0.9% Myrrh oil. Mean ± SD with different superscripts (small alphabet a, b, c, d, e, f, and g) differ significantly (*p* < 0.5).

**Table 1 molecules-26-06853-t001:** Content compounds identified (%) of *Eucalyptus* and Myrrh oil.

	Myrrh Oil		*Eucalyptus* Oil	
	Compound	Percentage (%)	Compound	Percentage %
1	δ-elemene	2.2	α-pinene	24.83
2	β-elemene	6.7	1,8-cineol	47.8
3	Germacrene B	4.5	Terpineol alpha	5.55
4	γ-elemene	1.7	pinocarveol	3.42
5	Germacrene D	2.9	4-terpineol	1.66
6	isofuranogermacrene	18.4	Trans-Pinocarveol	7.34
7	T-cadinol	2.9	alpha-terpinyl acetate	1.54
8	Furanodiene	21.3	Limonene	1.2
9	Furanoeudesma-1,3-diene	23.4	Pinacarvone	4.11
10	Lindestrene	7.5	Guaiene	1.53
11	2-methoxyfuranodiene	2.3	Spathulenol	1.02
12	2-acetoxyfuranodiene	6.2		

**Table 2 molecules-26-06853-t002:** The average scores for sensory evaluation of plain and fortified yogurt with *Eucalyptus* and Myrrh oil.

Treatment	Storage Time	Appearance	Mouthfeel	Texture	Flavor	Overall Acceptability
**C**	D1	9.6 ± 0.10 ^A^	9.2 ± 0.10 ^A^	9.4 ± 0.10 ^A^	9.3 ± 0.10 ^A^	9.4 ^A^
	D 14	9.4 ± 0.10 ^AB^	9.2 ± 0.10 ^A^	9.2 ± 0.10 ^A^	9.1 ± 0.09 ^AB^	9.2 ^B^
**E1**	D1	9.8 ± 0.10 ^A^	9.6 ± 0.10^A^	9.4 ± 0.10 ^A^	9.5 ± 0.10 ^A^	9.6 ^A^
	D 14	9.6 ± 0.10 ^AB^	9.4 ± 0.10 ^A^	9.6 ± 0.10 ^A^	9.3 ± 0.09 ^AB^	9.5 ^A^
**E2**	D1	9.4 ± 0.09 ^AB^	8.9 ± 0.09 ^A^	8.9 ± 0.09 ^A^	8.7 ± 0.09 ^AB^	9.0 ^C^
	D 14	8.8 ± 0.09 ^AB^	8.4 ± 0.08 ^A^	9.1 ± 0.10 ^A^	8.5 ± 0.08 ^B^	8.7 ^C^
**E3**	D1	8.7 ± 0.09 ^AB^	8.5 ± 0.09 ^A^	9.3 ± 0.10 ^A^	8.6 ± 0.09 ^AB^	8.8 ^C^
	D 14	8.5 ± 0.09^AB^	8.4 ± 0.08 ^A^	9.2 ± 0.09 ^A^	8.5 ± 0.09 ^AB^	8.7 ^C^
**M1**	D1	9.0 ± 0.09 ^AB^	8.6 ± 0.09 ^A^	8.6 ± 0.09 ^A^	8.4 ± 0.09 ^AB^	88 ^C^
	D 14	8.2 ± 0.09 ^AB^	8.1 ± 0.08 ^A^	8.9 ± 0.10 ^A^	8.1 ± 0.08^B^	8.4 ^E^
**M2**	D1	8.2 ± 0.09 ^AB^	8.2 ± 0.09 ^A^	9.2 ± 0.10 ^A^	8.3 ± 0.09 ^AB^	8.6 ^D^
	D 14	8.2 ± 0.09 ^AB^	8.0 ± 0.08 ^A^	8.8 ± 0.09 ^A^	8.2 ± 0.09 ^AB^	8.4 ^E^
**M3**	D1	8.6 ± 0.09 ^AB^	8.8 ± 0.09 ^A^	9.1 ± 0.09 ^A^	8.3 ± 0.09 ^AB^	8.8 ^C^
	D 14	8.0 ± 0.08 ^B^	8.4 ± 0.09 ^A^	8.8 ± 0.09 ^A^	8.2 ± 0.09 ^AB^	8.4 ^E^

C: plain yogurt, E1: yogurt fortified with 0.3% *Eucalyptus* oil, E2: yogurt fortified with 0.6%, *Eucalyptus* oil, E3: yogurt fortified with 0.9% *Eucalyptus* oil, M1: yogurt fortified with 0.3% Myrrh oil, M2: yogurt fortified with 0.6%, Myrrh oil, M3: yogurt fortified with 0.9% Myrrh oil. Values are mean ± SD of three independent replicates. Means with different superscripts (^A, AB, B, C, D^ and ^E^) are significantly different (*p* < 0.05).

**Table 3 molecules-26-06853-t003:** Total solids (TS) %, pH, and Syneresis volume (%) of yogurt samples during storage at 4 °C.

Treatment	TS (%)			pH			Syneresis (%)		
	d 1	d 7	d 14	d 1	d 7	d 14	d 1	d 7	d 14
**C**	16.8 ± 0.32 ^ab^	17.1 ± 0.75^a^	17.2 ± 0.61 ^a^	4.55 ± 0.1 ^b^	4.43 ± 0.14 ^b^	4.42 ± 0.13 ^b^	9.84 ± 0.62 ^f^	11.33 ± 0.71 ^b^	12.43 ± 0.78 ^a^
**E1**	17.1 ± 0.02 ^a^	17.3 ± 0.53 ^a^	17.5 ± 0.51 ^a^	4.64 ± 0.1 ^a^	4.56 ± 0.1 ^b^	4.55 ± 0.13 ^b^	8.72 ± 0.5 ^j^	9.61 ± 0.60 ^g^	10.83 ± 0.68 ^b^
**E2**	17.3 ± 0.64 ^a^	17.4 ± 0.42 ^a^	17.6 ± 0.61 ^a^	4.68 ± 0.15 ^a^	4.63 ± 0.14 ^a^	4.61 ± 0.14^a^	8.65 ± 0.52 ^k^	8.66 ± 0.54 ^j^	10.2 ± 0.63 ^e^
**E3**	17.2 ± 0.33 ^a^	17.4 ± 0.54 ^ac^	17.5 ± 0.12 ^a^	4.70 ± 0.15 ^a^	4.58 ± 0.14 ^b^	4.53 ± 0.14 ^b^	8.44 ± 0.55 ^h^	9.50 ± 0.60 ^e^	9.75 ± 0.68 ^e^
**M1**	17.0 ± 0.42 ^ab^	17.2 ± 0.42 ^a^	17.3 ± 0.41 ^a^	4.58 ± 0.15 ^b^	4.47 ± 0.14 ^b^	4.44 ± 0.13 ^b^	9.12 ± 0.58 ^h^	9.89 ± 0.62 ^ef^	11.33 ± 0.71 ^b^
**M2**	17.0 ± 0.42 ^a^	17.2 ± 0.52 ^a^	17.5 ± 0.61 ^a^	4.64 ± 0.15 ^a^	4.55 ± 0.15 ^b^	4.55 ± 0.14 ^b^	8.83 ± 0.57 ^i^	9.51 ± 0.60 ^g^	10.66 ± 0.67 ^d^
**M3**	17.1 ± 0.42 ^a^	17.3 ± 0.51 ^a^	17.7 ± 0.81 ^a^	4.66 ± 0.15 ^a^	4.56 ± 0.15 ^b^	4.50 ± 0.14 ^b^	8.12 ± 23 ^h^	9.14 ± 21 ^e^	9.84 ± 51 ^e^

C: plain yogurt, E1: yogurt fortified with 0.3% *Eucalyptus* oil, E2: yogurt fortified with 0.6%, *Eucalyptus* oil, E3: yogurt fortified with 0.9% *Eucalyptus* oil, M1: yogurt fortified with 0.3% Myrrh oil, M2: yogurt fortified with 0.6%, Myrrh oil, M3: yogurt fortified with 0.9% Myrrh oil. Values are mean ± SD of three independent replicates. Means with different superscripts (small alphabet ^a, b, c, d, e, f, g, h, i, ab, ac^ and ^ef^) are significantly different (*p* < 0.05).

**Table 4 molecules-26-06853-t004:** Total phenols (µg Gallic acid equivalent/g sample) & Antioxidant potential measured by DPPH radical scavenging activity.

Treatment	Total Phenolic Content			Antioxidant Activity		
	d 1	d 7	d 14	d 1	d 7	d 14
**C**	7.12 ± 0.11 ^k^	6.22 ± 0.09 ^k^	5.12 ± 0.08 ^l^	15.33 ± 0.16 ^i^	14.44 ± 0.14 ^j^	13.82 ± 0.14 ^k^
**E1**	20.47 ± 0.31 ^f^	17.16 ± 0.26 ^h^	15.45 ± 0.24 ^i^	52.33 ± 0.52 ^f^	49.1 ± 0.49 ^g^	45.11 ± 0.45 ^h^
**E2**	28.60 ± 0.44 ^b^	25.48 ± 0.39 ^c^	23.58 ± 0.36 ^d^	73.12 ± 0.73 ^a^	62.04 ± 0.62 ^d^	56.84 ± 0.57 ^e^
**E3**	30.37 ± 0.34 ^a^	28.36 ± 0.28 ^b^	26.35 ± 0.25 ^c^	75.58 ± 0.76 ^a^	72.9 ± 0.73 ^b^	67.45 ± 0.67 ^b^
**M1**	18.21 ± 0.32 ^g^	16.32 ± 0.53 ^h^	14.31 ± 0.34 ^i^	50.11 ± 0.55 ^f^	46.21 ± 0.39 ^g^	43.72 ± 0.23 ^h^
**M2**	26.21 ± 0.21 ^c^	23.32 ± 0.21 ^d^	22.63 ± 0.32 ^f^	69.21 ± 0.13 ^b^	59.92 ± 0.33 ^d^	53.21 ± 0.32 ^e^
**M3**	28.53 ± 0.34 ^b^	26.22 ± 0.21 ^c^	25.33 ± 0.35 ^c^	71.21 ± 0.36 ^a^	69.01 ± 0.21 ^b^	63.21 ± 0.22 ^c^

C: plain yogurt, E1: yogurt fortified with 0.3% *Eucalyptus* oil, E2: yogurt fortified with 0.6%, *Eucalyptus* oil, E3: yogurt fortified with 0.9% *Eucalyptus* oil, M1: yogurt fortified with 0.3% Myrrh oil, M2: yogurt fortified with 0.6%, Myrrh oil, M3: yogurt fortified with 0.9% Myrrh oil. Values are mean ± SD of three independent replicates. Means with different superscripts (small alphabet ^a, b, c, d, e, f, g, h, i, j, k^ and ^l^) are significantly different (*p* < 0.05).

## Data Availability

Not applicable.

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
