# Peer review of "Buffalo Yogurt Fortified with Eucalyptus (Eucalyptus camaldulensis) and Myrrh (Commiphora Myrrha) Essential Oils: New Insights into the Functional Properties and Extended Shelf Life"

_molecules, 2021, doi:10.3390/molecules26226853_

Round 1

Reviewer 1 Report

The authors propose a manuscript titled “Buffalo yogurt fortified with Eucalyptus and Myrrh essential oils: new insights into the functional properties and extended shelf life”

The article is original and well structured. In particular, this study takes into consideration the essential oils (EOs) of species of genera Eucalyptus and Myrrh and their benefits in terms of health and functionality. In particular was investigated the Buffalo set yogurt enriched with different concentrations of EOs, his relative antibacterial activity and bioactive properties. The results show the best organoleptic properties of yogurt are those coming samples treated with eucalyptus oil, and that the antioxidant activity, antibacterial activity, and total phenolic content were enhanced by increasing the concentration of EOs. Very interesting data is that the yogurt with eucalyptus oil showed the highest antioxidant activity and the highest antibacterial activity against S. typhimurium, while the highest antibacterial effect against L. monocytogene is that for myrrh oil-enriched yogurt. The authors conclude that the results show that eucalyptus and myrrh oils can be applied to yogurt to improve its beneficial properties on physical characteristics and human health due to their antioxidant activity and phenolic materials.

I read the work carefully in a critical way, and suggested in detail some crucial concepts only in order to further improve the work done. In particular, some concepts need to be referenced, indicating point by point where is necessary. Is also necessary introduce, with two word as I suggested, about the buffalo grazing, the management and others. Finally in a interational cotext, as Molecules, it is not ammissible to producing a table of essential oils without mentioning the botanical species considered, but only a genus

Abstract. Please summarize, not report a specific data with percentage ….

  1. Introduction

Well done, the concept are correct but in some case need to complete in the correct way. The suggestions are in bold:

  • Lines 26-27. Functional yogurt plays an important role in human nutrition due to its content of proteins, lactose, calcium, and water-soluble vitamins [choose a reference];
  • Lines 28-30. “Over the last few years, some functional milk products have been produced using fortification and enrichment, as consumers need good taste nutritious foods [choose a reference]”;
  • Lines 31-34. “To enhance the quality of newly designed functional food products, food fortification is defined by the inclusion of one or more components regardless of their natural presence on food (since that they occur naturally in food products) [choose a reference]”;
  • Lines 37-38. Listeria monocytogenes (Murray et al., 1926) Pirie, 1940 instead Listeria monocytogenes (L. monocytogenes). Remember when cited for the first time in the manuscript the plant or animal scientific name you must reporting the complete name with the author. ..both for species and genus (scientific name in italic).
  • Line 39. Commiphora myrrha (T. Nees) Engl. (common name: Myrrh) comes from Burseraceae family instead “Myrrh comes from the Commiphora trees family of Burseraceae [5];
  • Line 51. Please clarify, I don’t understand “The main advantage of myrrh is that it is equally botanical and herbal”;
  • Line 54. Different species of genus Eucalyptus L'Hér. instead Different species of eucalyptus. The correct way for botanical point of view;
  • Line 58. Eucalyptus in italic; check whole document and correct in the suggested way;
  • Lines 60-61. Eucalyptus. The first letter of genus always in uppercase;
  • Line 61. microbial strains and species instead microbial strains and microbial species
  • Line 62. Focal point. Please introduce with two words the buffalo as done for Eucalyptus and Commiphora, ….which species of Buffalo, management, the importance of Buffalo pasture at global level…I suggest to add and complete this period: “In natural habitat the buffalo grazing is a determinant of soil and vegetation that can affect the structure and function of vegetation coverage in many ways [Noy-Meir et al. 1989], for which it is necessary to establish a grazing plan [Perrino et al. 2021, Bell et al. 2013], a crucial manegement actions which have affects on antioxidant activity and chimcal composition of the final product, as yogurt and milk (Choose a reference), while in a confined environment play a crucial role the food provided by farmers….(please continue the concept, see and read literature”

References to be added

  • Noy-Meir, I.; Gutman, M.; Kaplan, Y. Responses of Mediterranean grassland plants to grazing and protection. Ecol. 1989, 77, 290–310.
  • Perrino, E.V.; Magazzini, P.; Musarella, C.M. Management of grazing “buffalo” to preserve habitats by Directive 92/43 EEC in a wetland protected area of the Mediterranean coast: PaludeFrattarolo, Apulia, Italy. Euro-Mediterr. J. Environ. Integr. 2021, 6, 32.
  • Bell, L.W.; Moore, A.D.; Kirkegaard, J.A. Evolution of crop–livestock integration systems that improve farm productivity and environmental performance in Australia. J. Agron. 2013, 57, 10–20.
  1. Materials and methods

There is a lack in the work. Is it important to know the origin of the buffalo milk, and therefore the type of farming adopted for the buffalo, i.e. free grazing in wetland areas?, grazing confined in a fence and artificially fed? This is crucial aspect because modify the basic antibacterial characteristics of the yogurt!!! (see my previous comment)

  • Lines 71-74. The authors statement: “Fresh buffalo milk used in yogurt manufacturing was obtained from the herd of Faculty of Agriculture, Cairo University, Giza, Egypt”. The buffalo milk come from grazing of Faculty of Agricolture? sure?.
  1. Results and Discussion

The figures and tables are clear, but there is no word as to which species the authors refer.

  • The result on essential oils refer generally to genera eucaliptus and myrrh, it’s not lawful in a scientific work. Please specify the species, with scientific name about genera Eucalyptus and Commiphora
  • In Eucalyptus did not find Eucalyptol compound?

Please follow the guidelines of the journal about references

Author Response

Dear Prof. Dr.

Thank you for your efforts, cooperation, and valuable assistance to researchers in the publication process. I send you the modified manuscript according to your vision and responding to your comments.

Comment

Response

Abstract. 

Please summarize, not report a specific data with percentage

Some sentences of the abstract has been summarized as commented by reviewer.

Lines 26-27. Functional yogurt plays an important role in human nutrition due to its content of proteins, lactose, calcium, and water-soluble vitamins [choose a reference];

The reference has been added number 1 [1] Ozturkoglu-Budak S; Akal C; Yetisemiyen A. Effect of dried nut fortification on functional, physicochemical, textural, and microbiological properties of yogurt. Journal of dairy science 2016, 99, 8511-8523.

Line 27

Lines 28-30. “Over the last few years, some functional milk products have been produced using fortification and enrichment, as consumers need good taste nutritious foods [choose a reference]”;

The reference has been added number 3 [3] Chadare FJ; Idohou R; Nago E; Affonfere M; Agossadou J; Fassinou TK; Kénou C; Honfo S; Azokpota P; Linnemann AR; Hounhouigan DJ. Conventional and food‐to‐food fortification: An appraisal of past practices and lessons learned. Food science & nutrition. 2019, 7, 2781-95.

Line 30

Lines 31-34. “To enhance the quality of newly designed functional food products, food fortification is defined by the inclusion of one or more components regardless of their natural presence on food (since that they occur naturally in food products) [choose a reference]”;

The reference has been added number 5 [5] Öztürk B. Nanoemulsions for food fortification with lipophilic vitamins: Production challenges, stability, and bioavailability. European Journal of Lipid Science and Technology. 2017, 119, 1-18.

Line 35

Lines 37-38. Listeria monocytogenes (Murray et al., 1926) Pirie, 1940 instead Listeria monocytogenes (L. monocytogenes). Remember when cited for the first time in the manuscript the plant or animal scientific name you must reporting the complete name with the author. ..both for species and genus (scientific name in italic).

The sentence added

Line 39

Line 39. Commiphora myrrha (T. Nees) Engl. (common name: Myrrh) comes from Burseraceae family instead “Myrrh comes from the Commiphora trees family of Burseraceae [5];

The comment has been done

Lines 41-42

Line 51. Please clarify, I don’t understand “The main advantage of myrrh is that it is equally botanical and herbal”;

The sentence has been modified

Lines 53 – 54.

Line 54. Different species of genus Eucalyptus L'Hér. instead, Different species of eucalyptus. The correct way for botanical point of view;

The comment has been done

lines 56-57

Line 58. Eucalyptus in italic; check whole document and correct in the suggested way;

Done in all text

Lines 60-61. Eucalyptus. The first letter of genus always in uppercase;

Done in all text

Line 61. microbial strains and species instead microbial strains and microbial species

The word has been changed

Line 64

Line 62. Focal point. Please introduce with two words the buffalo as done for Eucalyptus and Commiphora, ….which species of Buffalo, management, the importance of Buffalo pasture at global level…I suggest to add and complete this period: “In natural habitat the buffalo grazing is a determinant of soil and vegetation that can affect the structure and function of vegetation coverage in many ways [Noy-Meir et al. 1989], for which it is necessary to establish a grazing plan [Perrino et al. 2021, Bell et al. 2013], a crucial manegement actions which have affects on antioxidant activity and chimcal composition of the final product, as yogurt and milk (Choose a reference), while in a confined environment play a crucial role the food provided by farmers….(please continue the concept, see and read literature”

The paragraph has been added and also the following references were added

[20] Noy-Meir I; Gutman M; Kaplan Y. Responses of Mediterranean grassland plants to grazing and protection. Ecol. 1989, 77, 290–310.

[21] Perrino EV; Magazzini P; Musarella CM. Management of grazing “buffalo” to preserve habitats by Directive 92/43 EEC in a wetland protected area of the Mediterranean coast: PaludeFrattarolo, Apulia, Italy. Euro-Mediterranean Journal for Environmental Integration. 2021, 6, 32.

[22] Bell LW; Moore A.D; Kirkegaard JA. Evolution of crop–livestock integration systems that improve farm productivity and environmental performance in Australia. J. Agron. 2013, 57, 10–20.

[23] Sraïri M.T; Benhouda H; Kuper, M; Le Gal PY. Effect of cattle management practices on raw milk quality on farms operating in a two-stage dairy chain. Tropical Animal Health and Production, 2009, 41, 259–272.

Lines 64 -72

Materials and methods

Lines 71-74. The authors statement: “Fresh buffalo milk used in yogurt manufacturing was obtained from the herd of Faculty of Agriculture, Cairo University, Giza, Egypt”. The buffalo milk come from grazing of Faculty of Agriculture? sure?

Yes, the buffalo milk comes from the herd of Faculty of Agriculture, Cairo University I am sure, because I was bringing the milk by myself.

Results and Discussion

The result on essential oils refer generally to genera eucaliptus and myrrh, it’s not lawful in a scientific work. Please specify the species, with scientific name about genera Eucalyptus and Commiphora

The scientific name about genera Eucalyptus and Commiphora were added in the title and abstract

In Eucalyptus did not find Eucalyptol compound?

I have obtained these data from the lab. Maybe Eucalyptol not detected by the instrument or not found in the library.

Please follow the guidelines of the journal about references

The guidelines of the journal have been followed about references

Reviewer 2 Report

I have identified many English mistakes in the manuscripts: I recommend to the authors to have they paper reviewed by an English native.

Abstract:

Line 18: close the parenthesis.

Introduction

The authors should define what a functional food is. Line 32.

Please homogenize the abbreviation of Essential oils (EOs or Eos) in the text. Line 34

This sentence should come earlier in the text. Line 41-42

I think that authors should cite the name of the first author because beginning by a number is quite disrupting. Line 47.

Go to the line after [11] line 54

The authors did not work on human health and this point cannot be mentioned has an objective in the end of the introduction. Line 63.

Again the authors should define what is a functional properties and a functional food product? Line 66

Materials and methods

The authors should specify the starter culture used, name and composition...Line 67

The authors should give more details concerning the composition of the milk Line 86. (Protein content for example that his very important for the yoghurt and also the water content of the milk...).

Precision should be given concerning the heat treatment (process) applied to the milk. Line 87

The authors should explain why those three concentrations were used. Line 90.

If the milk is homogenized, the authors should give details concerning the process of homogenization used. Why the EOs where added before homogenization and not after. This can induce some loss of the EOs added. Line 92.

How the culture was added. It is important to specify how the started was prepared. Its initial concentration... Line 93.

How the pH was controlled Line 94.

The authors should also specify the container of the yogurts. Line 94.

Section 2.2.5. Please precise the number of panelists that tested the product. Are they trained ones or not?  How the samples were presented... More details should be given to understand clearly the methodology used.

What kind of analysis were performed on the supernatant?  Line 119.

Please precise the device used. Line 125 and 135.

Part 2.2.6 and 2.2.9 should be merged for more clarity.

2.2.10. Please precise the software used for performing data analysis.

General comment: I think that some paragraphs of the material and method section should be reorganized in order to facilitate the reading and understanding of the experiments. This will prevent the reader from having questions and having to wait to move on to another paragraph in this section in order to understand how the experiments were conducted.

Results and discussions

Line 154 to 159. Revise the English. Is there any difference between the study and the one cited line 157? Please compare clearly your results to the literature.

Lie 171-173. Here the two types of yogurts should be compared clearly concerning there sensory properties.

Line 190. Not clear. Please reformulate.

What is the mining of TS please define in the text and in the table 3.

Line 191. What is Chamaemelum oil? This was not defined previously and why the authors compared they results to this kind of oil.

Title of table 3. Correct the title of table 3 because the samples were also analysed at days 1 and 7 .

Line 211. Please correct the table number.

Line 212. The values are not correct the data range is from 9.84 to 12.43 for the control (table 3).

For the figure 1, I suggest to the authors to use histogram and not a curve to represent there results. This will be clearer for the reader. Moreover, the authors should also specify the mining of the letter c, g, e, d... present in the figure. They also need to add title of the y - axis.

Conclusion. The first sentence of the conclusion must be reformulated because this one is not clear enough and is not related to the results and conclusions formulated in the manuscript.

Author Response

Dear Prof. Dr.

Thank you for your efforts, cooperation, and valuable assistance to researchers in the publication process. I send you the modified manuscript according to your vision and responding to your comments.

Comment

Response

I have identified many English mistakes in the manuscripts: I recommend to the authors to have they paper reviewed by an English native.

The manuscript has been reviewed by Prof. Dr. Sanaa Badran who is an English native.

Abstract:

Line 18: close the parenthesis.

The parenthesis has been closed

Line 19

Introduction

The authors should define what a functional food is. Line 32.

The definition of functional food products has been added

Lines 32 -33.

Please homogenize the abbreviation of Essential oils (EOs or Eos) in the text. Line 34

The abbreviation of Essential oils (EOs or Eos) has been homogenized in all the text as (EOs)

This sentence should come earlier in the text. Line 41-42

The position of the sentence has been changed

Line 40-41

I think that authors should cite the name of the first author because beginning by a number is quite disrupting. Line 47.

The name of the first author has been cited Line 49

The authors did not work on human health and this point cannot be mentioned has an objective in the end of the introduction. Line 63.

The words “human health” has been deleted

Again the authors should define what is a functional properties and a functional food product? Line 66

The definition of functional food products has been added

Lines 32 -33.

Materials and methods

The authors should specify the starter culture used, name and composition...Line 67

The starter culture name and composition used were added

Lines 82 – 84.

The authors should give more details concerning the composition of the milk Line 86. (Protein content for example that his very important for the yoghurt and the water content of the milk...).

The chemical composition of milk has been added

Lines 98-99.

Precision should be given concerning the heat treatment (process) applied to the milk. Line 87

The heat treatment was controlled carefully during the process using calibrated thermometer.

The authors should explain why those three concentrations were used. Line 90.

The reason for using these three concentrations were explained as follow

“These three concentrations were used after manufacturing of the different formulations of yoghurt and a preliminary sensory evaluation study was conducted to select the best formulations that will be used throughout the experiment”

Lines 107-109.

If the milk is homogenized, the authors should give details concerning the process of homogenization used. Why the EOs where added before homogenization and not after. This can induce some loss of the EOs added. Line 92.

Details about the homogenization process were added

Homogenization of all the batches was carried out at 60 °C in a two-stage homogenizer (M/S Goma Engineers, Mumbai) with 1,000 and 500 psi pressures at first and second stages, respectively.

Adding essential oils after homogenization to prevents the oiling off during fermentation and storage.

Lines 99 – 101.

How the culture was added. It is important to specify how the started was prepared. Its initial concentration... Line 93.

Adding culture were written

Starter cultures were prepared by cultivated in 10% nonfat dry milk, autoclaved at 115 °C for 15 min, inoculated under sterile conditions, incubated at 42 °C and kept in a refrigerator at 4 °C until used. Then, the inoculated milk incubated at 42°C

Lines 112-114

How the pH was controlled Line 94.

Control of pH was by sampling either in the fermentation tanks or directly in cups, and duration of the different steps of manufacture.

Lines 114-116.

The authors should also specify the container of the yogurts. Line 94.

The yogurt was packaged in covered plastic cups and cooled down to 4-5 °C.

Lines 116-117.

Section 2.2.5. Please precise the number of panelists that tested the product. Are they trained ones or not?  How the samples were presented... More details should be given to understand clearly the methodology used.

Were illustrated as follow:

A random selection of yogurt samples was evaluated for Availability, surface smoothness, mouthfeel, and texture (soft, grain, springiness) flavor, and taste (sweetness, aroma) and overall admissibility by members of the department of dairy science of faculty of agriculture at Cairo University. All these trained panelists (15 person) experts and were chosen based on their desire to participate and their knowledge about dairy products. They were frequent yoghurt consumers and did not have any allergies to it. Plain- and for-tified-yoghurt samples were presented in plastic coded (three-digit random codes) cups. Each cup contains 50 mL of yoghurt samples that freshly removed from the refrigerator.

Lines 132 -139.

What kind of analysis were performed on the supernatant?  Line 119.

The analysis is antibacterial activity

Line 162.

Please precise the device used. Line 125 and 135.

The device name is UV/Vis. Spectrophotometer, Jenway. England.

Line 148.

Part 2.2.6 and 2.2.9 should be merged for more clarity.

Parts 2.2.6 and 2.2.9 were merged

“To obtain the yogurt supernatant, samples were centrifuged (Centrifuge version C-28 AC BOECO, Hamburg, Germany) at 4000 xg for 30 min at 4°C [20]. The supernatants obtained were filtered with the 0.45-μm diameter millipore membrane filter and stored at -20°C for further analysis (antibacterial activity). The supernatant filters were used to assess the antibacterial efficacy of certain pathogens (E. coli ATCC35218, L. monocytogenes ATCC19115, and S. typhimurium ATCC14028) according to [20] using agar well diffusion assay. A diluted sample of 100 μL of yogurt supernatant is distributed on agar plates. The inhibition zone was measured after incubation at 37 °C /24- 48 h. The bacterial activity was expressed as the inhibition zone (mm). Each test was conducted three times.

Lines 159-168

2.2.10. Please precise the software used for performing data analysis.

The software used for performing data analysis were MSTAT-C

Lines 170 – 171.

General comment: I think that some paragraphs of the material and method section should be reorganized in order to facilitate the reading and understanding of the experiments. This will prevent the reader from having questions and having to wait to move on to another paragraph in this section in order to understand how the experiments were conducted.

Some paragraphs of the material and method have been reorganized.

Results and discussions

Line 154 to 159. Revise the English. Is there any difference between the study and the one cited line 157? Please clearly compare your results to the literature.

The English has been revised and the comparison between the results and literature has been discussed

Lines 180-182

Results indicate a significant amount of the major constituents identified in the Myrrh oil (10.6, 7.4, 18.4, 23.4, and 6.2 %) for elements, Germacrene B and D, isofuranogermacrene, Furanodieneb, Furanoeudesma-1,3-diene, Lindestrene, and 2-acetoxyfuranodiene., respec-tively. Some previous studies reported by [33] found that the major compounds identified in Myrrh resin were 2-acetoxyfuranodiene (9.80%), furanoeudesma-,3-diene (8.97%), iso-furanoger macrene (6.71%), and epicurzerenone (3.643%), followed by 2- methoxyfurano-diene (2.97%) and lindestrene (2.74%). In the same context, Its main components, identi-fied and quantified by GC/MS, were furanoeudesma-1,3-diene, 34.9%; lindestrene, 12.9%; curzerene, 8.5%; and germacrone, 5.8% [34]. It was noted that there are some differences, whether in the proportion of Myrrh com-pounds or their types, this could be due to the type of plant or the environment.

Lines 183-191

Lie 171-173. Here the two types of yogurts should be compared clearly concerning there sensory properties.

The comparison between the two types of fortified yogurt were written .”However, the yogurt fortified with Eucalytus oil was more acceptable than yogurt fortified with Mrrh oil”.

Lines 207 -208.

Line 190. Not clear. Please reformulate.

The sentence has rewritten in lines 225 -226

What is the mining of TS please define in the text and in the table 3.

TS defined as (total solids) in the text and in the table 3.

Lines 224 and 239

Line 191. What is Chamaemelum oil? This was not defined previously and why the authors compared they results to this kind of oil.

The sentence has been deleted.

Title of table 3. Correct the title of table 3 because

the samples were also analysed at days 1 and 7 .

The title of table 3 has been corrected

Line 239

Line 211. Please correct the table number.

The table number has been corrected

Line 245

Line 212. The values are not correct the data range is from 9.84 to 12.43 for the control (table 3).

The values have been corrected

Line 246.

For the figure 1, I suggest to the authors to use histogram and not a curve to represent there results. This will be clearer for the reader. Moreover, the authors should also specify the mining of the letter c, g, e, d... present in the figure. They also need to add title of the y - axis.

The figure 1 has been changed to histogram instead of curve.

The titles of the y – axis have been added

the meaning of the letter has been present in the figure.

Mean ± SD with different superscripts (small alphabet a, b, c, d, e, f and g) differ significantly (p < .05)

Lines 316-317

Conclusion. The first sentence of the conclusion must be reformulated because this one is not clear enough and is not related to the results and conclusions formulated in the manuscript.

The first sentence has changed to: Fortifying yogurt with Essential oils is of great interest to improve the functionality and increase its shelf life

Lines 320-321

Round 2

Reviewer 1 Report

I appreciate the revisions done by the authors in the last version of the proposed manuscript. Now it can be published without other changes.

Reviewer 2 Report

Dear authors, thank you for all the responses to the various comments I made. However, the article is even more confusing than when it was first submitted. It is important to take the time to reorganize the text and the answers provided in the text. As it is, these answers do not bring any clarification because they are badly written or badly positioned in the text. 

Best regards.